# Gender differences in dry eye disease symptoms associated with psychological health indicators among adults using mobile mental health apps

Hyojin Kim[1]◉, Youngju An[2]◉, Won Ju Hwang[3]*

1 Department of Optometry, Division of Health Science, and Graduate School of Health and Welfare, Baekseok University, Cheonan, Korea, 2 Department of Optometry, Baekseok Culture University, Cheonan, Korea, 3 Department of College of Nursing Science, East-west Nursing Research Institute, Kyung Hee University, Seoul, Korea

◉ These authors contributed equally to this work.
* hwangwj@khu.ac.kr

**Data Availability Statement:** All relevant data are within the paper and Supporting information files.

**Funding:** This research was supported by a grant from the Korea Health Technology R&D Project

## Abstract

### Purpose

We aimed to investigate the association between dry eye disease (DED) symptoms and mental health among Korean adults in the community.

### Methods

A cross-sectional study analyzed the data of 152 participants using mobile-phone applications for recording mental health. We defined individuals with DED symptoms as those who experienced a sense of irritation or dryness of the eye (either constantly or often). Mental health (perceived stress, depression, and anxiety) was assessed using the Perceived Stress Scale, Patient Health Questionnaire-9, and General Anxiety Disorder-7, in this order. Multiple logistic regression analysis was conducted to examine the association between DED symptoms and mental health. We also adjusted for possible covariates. We investigated sex differences in mental health status in relation to DED.

### Results

We found that 41.4% of the participants (48.4% female and 30.5% male) showed DED symptoms, and its prevalence was higher in female than in male. The average perceived stress and anxiety symptoms in the female with DED (24.69±4.73 and 6.56±5.09, respectively) were significantly higher than those without DED (21.38±4.68 and 4.54±4.03) ($p$ = 0.020 and 0.038, respectively). Even after adjusting for possible covariates, female who had DED showed higher risks of perceived stress (OR = 1.28), depression (OR = 2.64), and anxiety (OR = 5.81).

through the Korea Health Industry Development Institute (KHIDI), and it was funded by the Ministry of Health & Welfare, Republic of Korea (grant number: HI18C1317). The funding agencies had no role in the study design, the collection, analysis, or interpretation of data, the writing of the report, or the decision to submit the article for publication.

**Competing interests:** No.

## Conclusions

Clinicians and nurses should therefore be aware of the possibility of mental health problems among female with DED.

## Introduction

Dry eye disease (DED) is a major public health problem characterized by a deficiency in the quality of life in the sphere of ophthalmology [1]. The global prevalence rate of DED is reported as 4.3~52.4% [2–4], and 8–14.4% in Korea [5].

The Tear Film & Ocular Surface Society (TFOS), launched in the Dry Eye Workshop II (TFOS DEWS II), defines DED as a multifactorial disease affecting the secretion of tears and the ocular surface, accompanied by inflammation in the tear film affecting osmolality and the ocular surface [6]. DED results in symptoms such as dryness, pain, ocular irritation, foreign body sensation, and visual disturbances, which negatively affect the quality of life [7].

The importance of sex disparities in medical research has also emerged recently. Since differences between sexes with regard to ophthalmic diseases have been reported, it has become a major issue. According to previous studies, refractive errors (56.0%), age-related macular degeneration (65.0%), cataracts (61.0%), low vision (63.0%), blindness (66.0%) have higher occurrence rates in female than in male. DED is also known to have a higher occurrence in female than in male [5, 8, 9].

In line with this trend, the TFOS DEWS II report on gender, and hormones [9] argued that further studies are needed to clarify the exact nature and extent of sex effects on DED. Indeed, it has been reported that biological differences between sexes have significant effects on controlling the ocular surface and adnexa [9, 10]. Several studies have suggested that decreased androgen levels and exogenous estrogen use may be risk factors for DED in female [11]. In this context, it was found that an imbalance in sex steroids experienced by postmenopausal female can affect the meibomian glands [12].

According to a large-scale Women's Health Study, women are diagnosed with DED six years earlier than men on average [13]. Therefore, the quality of life of women is more affected because of early confirmation of DED. However, previous studies on mental health and DED showed limitations in that they did not consider sex differences as an important factor [1, 14]. Thus, this study focuses on sex differences in the mental health of DED patients.

## Material and methods

We introduced and promoted the study by posting online promotions and posters at communities (schools, companies, organizations, etc.), and those who agreed to participate in the research were included, and the purpose and method of the research were explained to them.

We initially included 202 participants who had completed the healing app program for analysis. Sixteen subjects aged < 19 or > 60 years were excluded. To avoid a selection bias as much as possible, the following measures were applied: (i) age over 60 has been demonstrated to be a risk factor for dry eye symptoms [15]; (ii) health app users over the age of 60 have been found to have low self-efficacy [16]. Of the 186 subjects, we excluded those who lacked information on dry eyes, could not participate in the mental health interviews and were a mismatch for the other covariates. Among the remaining 176 subjects, since previous studies indicated that patients taking antidepressants had a higher chance of developing DED [17], we

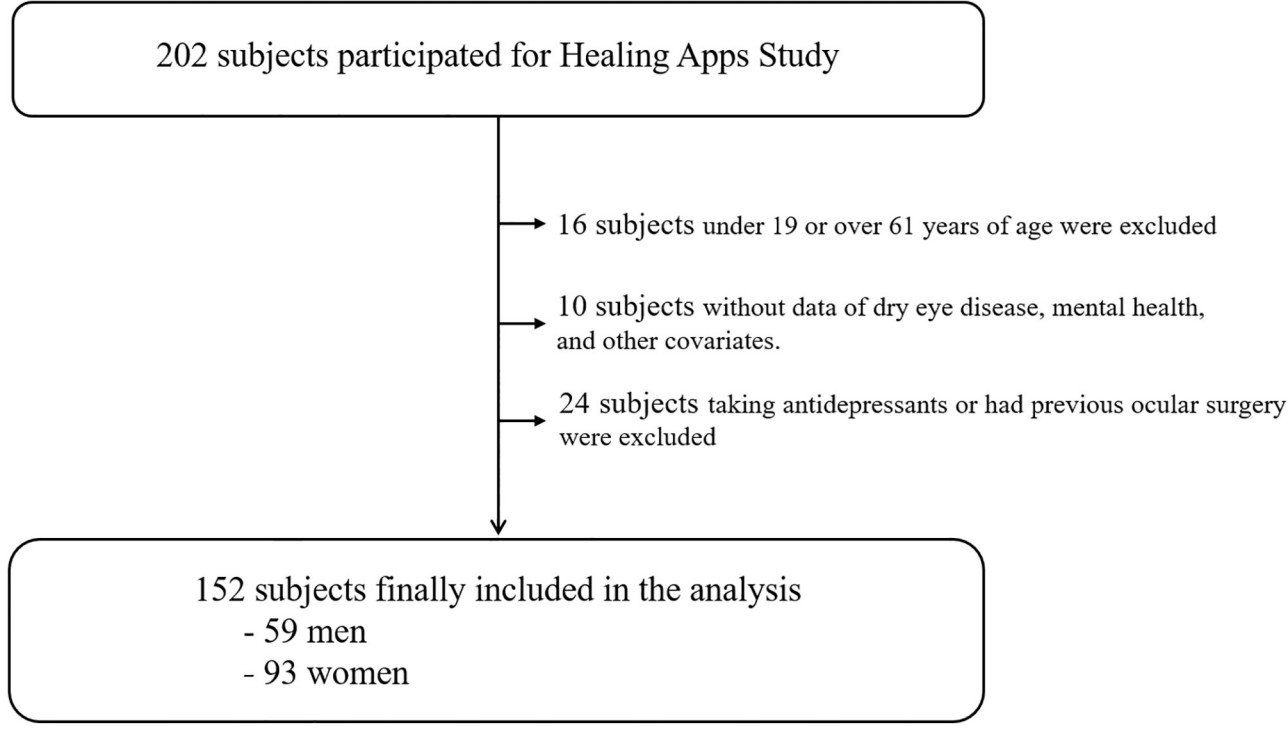

**Fig 1. Subject selection for final analysis.**

additionally excluded 24 subjects who were receiving treatment for depression or had previously undergone ophthalmic surgery. Finally, 152 subjects (59 male and 93 female) were included in the analysis (Fig 1).

Participants were to use a healing program, in the form of a mobile phone application, developed for promoting adult mental health (Fig 2). As a primary study, this study was first aimed at examining the participants' mental health and dry eye symptoms. The study was carried out after approval of the institutional review board of K University (KHSIRB-20-192 (EA)). In the future, we plan to have participants use the healing program on a regular basis through a mobile app to investigate the intervention effect. We conducted a questionnaire on participants in a healing program. An alarm was set to encourage participants to use the app; we induced continued use by providing health education every week. A description of the mobile mental health applications used is provided in the previous literature [18].

Questions such as the following, regarding the presence of DED symptoms were to be answered with "yes" or "no.": "Until now, have you ever had symptoms of DED before? For example, a sense of irritation or dryness of the eyes?" [19, 20]. Asking about associated DED symptoms is a reliable diagnostic method, as there are currently no definitive clinical diagnostic tests to diagnose DED [21, 22]. Previous studies have used this method to identify DED symptoms [19, 20].

Mental health was evaluated from the following three perspectives: perceived stress, depression, and anxiety. Stress perception was evaluated using the Perceived Stress Scale (PSS). The PSS was developed by Cohen et al. [17] and is known as the most common tool to check the cognition of subjective stress. The current study used the Korean version of the PSS (PS-10), which included 10 questions (6 about negative cognition, and 4 about positive cognition) The number of stressful situations the participant was in over the past month, was measured on a

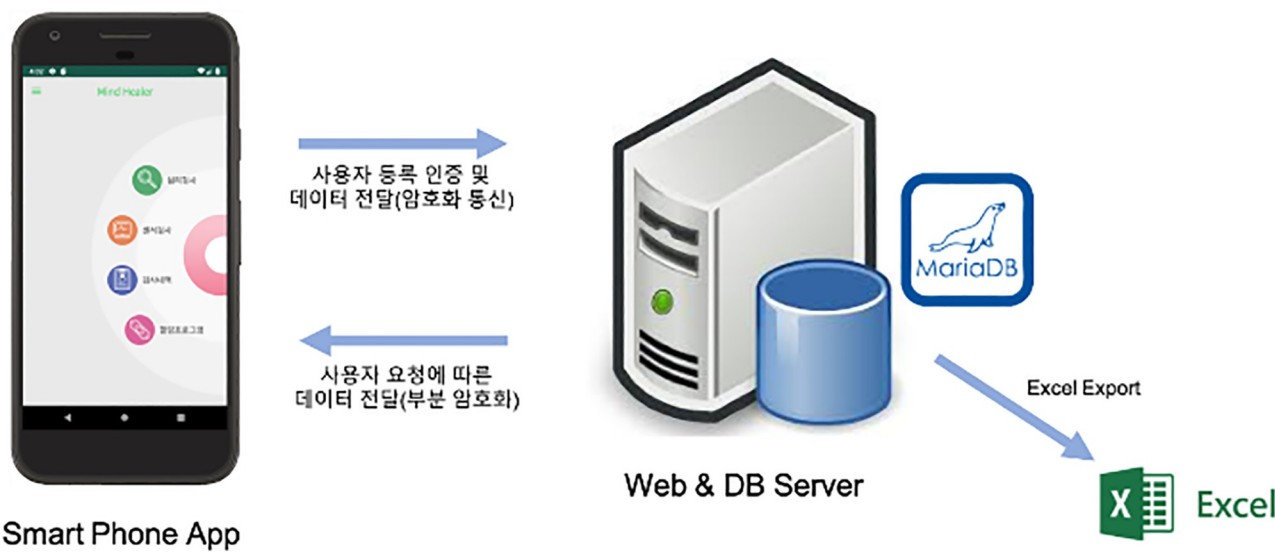

**Fig 2. Configuration of healing program.**

scale of 0 to 4 (5 scales). Higher scales indicated higher stress perception and were classified into three groups (low, middle, and high).

Depressive symptoms were evaluated using the Patient Health Questionnaire-9 (PHQ-9). This study uses the Korean version of PHQ-9 modified by Lee et al. [23], originally developed by Kroenke et al. [24]. A total of 9 questions regarding depression experiences in the past two months, on a scale of 0–27 was asked.

Higher scales indicated more depressive symptoms and were classified into three groups (low, middle, and high).

Anxiety symptoms were evaluated using the General Anxiety Disorder-7 (GAD-7). The GAD-7 includes seven questions regarding anxiety over the past two months on a scale of 0–21. Higher scales indicated higher levels of anxiety and were classified into three groups (low, middle, and high).

Questionnaires were used to collect data on age, sex, education, household income, occupation, marital status, body mass index, medicine, history of ocular surgery, and wearing contact lenses. Sex was classified as "male" and "female." Education included "high school" and "university or higher." Household income was classified as "lowest quartile," "2nd quartile," "3rd quartile," and "highest quartile." Occupation was categorized as "office worker" and "others.". Marriage status as "yes" or "never." BMI was calculated as the individual's weight in kilograms divided by the square of their height in meters. They were categorized as "underweight" if BMI<18.5, "normal" if between 18.5–25.0, and "obese" if BMI ≥25. Medicine was answered with "yes" or "no," to the question "Are you currently taking any medicine?" History of ocular surgery was to be answered with "yes" or "no," to "Have you previously undergone an ophthalmological surgery?" Wearing contact lenses was to be answered with "yes" or "no," to "Are you currently wearing contact lenses (including hard contact lenses)?"

Data are expressed as numbers (%) or means ±standard deviations. Unpaired t-tests and chi-square tests were used to compare the demographic, lifestyle, and clinical characteristics of the study population between male and female. Furthermore, an unpaired t-test was conducted to compare the total scores of mental health, including PSS, PHQ-9, and GAD-7, in relation to DED in male and female. The chi-square test was used to test the differences in the

distribution of mental health severity for statistical significance between male and female with and without DED. Multiple logistic regression analysis was conducted to examine the odds ratio (OR) and 95% confidence interval (CI) for the association between mental health related characteristics and DED in male and female. Demographic factors (age and sex), lifestyle factors (education, household income, marital status, occupation, and body mass index), and medical factors (medicine, history of ocular surgery, and wearing contact lenses) were used as covariates to calculate the adjusted odds ratios (aOR). Statistical analyses were performed using SPSS 18.0 version (SPSS Inc., Chicago, IL, USA). All reported p values which were 2-tailed and less than 0.050 were considered statistically significant.

## Results

A total of 152 participants were studied: 63 with no DED (41.4%) and 89 with DED (58.6%). We found that 41.4% (63) of the participants [48.4% (45/93) female and 30.5% (18/59) male] had DED, and the prevalence of DED was higher in female than in male.

Table 1 summarizes the evaluated parameters, including the demographic, lifestyle, and clinical characteristics of the 59 male and 93 female. The ages of the male (34.0±8.5yrs) and the female (34.4±9.4yrs, $p = 0.806$) did not differ significantly. In contrast, the proportion of female with DED symptoms was significantly higher than that of male ($p = 0.042$). Regarding mental health, the total PSS score was significantly higher in female than in male ($p = 0.001$).

Table 2 shows a comparison of mental health related variables (stress perception; PSS, depression symptom; PHQ-9, anxiety symptom; GAD-7) in relation to those who do, or do not have DED between male and female. Of the mental health variables, the total scores of PSS ($p = 0.020$) and GAD-7 ($p = 0.038$) in the group with DED (24.69±4.73 and 6.56±5.09) were significantly higher than those in the group without DED (21.38±4.68 and 4.54±4.03) in the female. In contrast, the scores of mental health related variables did not differ significantly between those who have and those who do not have DED among male. The average score of mental health related variables in relation to DED for both male and female are shown in Fig 3. In particular, the mental health related variables showed the highest score in female having DED symptoms.

Furthermore, the percentages of low, middle, and high scores for the mental health-related variables based on the participants' sex and DED group are shown in Fig 4. Female with DED symptoms had a higher proportion of female with higher mental health-related scores than those without. However, there was no significant difference in mental health-related scores based on the presence or absence of DED in male.

Table 3 shows the multivariate associations between mental health-related variables and DED in both male and female. Even after revision regarding demographic factors (age and sex), lifestyle factors (education, household income, married, occupation, body mass index), and medical factors (medicine, history of ocular surgery, and wearing contact lenses), mental health risks increased in those who had DED than those who did not [stress perception (OR = 1.12, 95% CI = 1.02–1.23), depressive symptoms (OR = 1.99, 95% CI = 1.02–3.90), and anxiety symptoms (OR = 3.50, 95% CI = 1.09–11.22)]. Analysis of sex differences showed no gap in male regarding the existence of DED, but female showed significant growth in mental health (psychological health indicators) risks. [Stress perception (OR = 1.28, 95% CI = 1.09–1.49), depressive symptoms (OR = 2.64, 95% CI = 1.10–6.35), and anxiety symptoms (OR = 5.81, 95% CI = 1.16–29.14)].

## Discussion

This study is the first to identify sex differences in mental health in subjects with DED symptoms. The main finding of this study was that in female, the association between DED

**Table 1. Evaluation parameters of male and female.**

| | Total (n = 152) | | Male (n = 59) | | Female (n = 93) | | *p*-value |
|---|---|---|---|---|---|---|---|
| | n | % | n | % | n | % | |
| Age (years) | 34.2±9.0 | | 34.0±8.5 | | 34.4±9.4 | | 0.806 [a] |
| Education | | | | | | | |
| High school | 23 | 15.1 | 10 | 16.9 | 13 | 14 | 0.647 [b] |
| University or higher | 129 | 84.9 | 49 | 83.1 | 80 | 86 | |
| Household income | | | | | | | |
| Lowest quartile | 43 | 28.3 | 17 | 28.8 | 26 | 28 | 0.751 [b] |
| 2nd quartile | 35 | 23 | 14 | 23.7 | 21 | 22.6 | |
| 3rd quartile | 35 | 23 | 11 | 18.6 | 24 | 25.8 | |
| Highest quartile | 39 | 25.7 | 17 | 28.8 | 22 | 23.7 | |
| Occupation | | | | | | | |
| Others | 115 | 75.7 | 45 | 76.3 | 70 | 75.3 | 1.000 [b] |
| Office worker | 37 | 24.3 | 14 | 23.7 | 23 | 24.7 | |
| Married | | | | | | | |
| Never | 87 | 57.2 | 38 | 64.4 | 49 | 52.7 | 0.180 [b] |
| Yes | 65 | 42.8 | 21 | 35.6 | 44 | 47.8 | |
| Body mass index | | | | | | | |
| Underweight | 14 | 9.2 | 52 | 88.1 | 78 | 83.9 | 0.063 [b] |
| Normal | 130 | 85.5 | 2 | 3.4 | 12 | 12.9 | |
| Obesity | 8 | 5.3 | 5 | 8.5 | 3 | 3.2 | |
| Medicine | | | | | | | |
| No | 145 | 95.4 | 59 | 100 | 86 | 92.5 | 0.043 [b] |
| Yes | 7 | 4.6 | 0 | 0 | 7 | 7.5 | |
| History of ocular surgery | | | | | | | |
| No | 123 | 80.9 | 47 | 79.7 | 76 | 81.7 | 0.833 [b] |
| Yes | 29 | 19.1 | 12 | 20.3 | 17 | 18.3 | |
| Wearing contact lenses | | | | | | | |
| No | 116 | 76.3 | 55 | 93.2 | 61 | 65.6 | <0.001 [b] |
| Yes | 36 | 23.7 | 4 | 6.8 | 32 | 34.4 | |
| Dry eye disease (DED) | | | | | | | |
| No | 89 | 58.6 | 41 | 69.5 | 48 | 51.6 | 0.042 [b] |
| Yes | 63 | 41.4 | 18 | 30.5 | 45 | 48.4 | |
| PSS | 21.4±5.29 | | 19.70±5.57 | | 22.50±4.82 | | 0.001 [a] |
| PHQ-9 | 6.75±4.76 | | 5.95±4.38 | | 7.26±4.94 | | 0.099 [a] |
| GAD-7 | 5.04±4.41 | | 4.29±3.90 | | 5.52±4.66 | | 0.094 [a] |

PSS, perceived stress scale; PHQ-9, patient health questionnaire-9; GAD-7, general anxiety disorder-7.

[a] Unpaired t-test

[b] Chi-square test

symptoms and mental health (stress perception, depression, and anxiety) showed a statistically significant association. However, no relationship was found between the presence of DED symptoms and mental health indicators in male.

The incidence of DED is well known to be higher in female than in male. Our study also showed that the proportion of female with DED symptoms was 48.4%, significantly higher than that of 30.5% of male (*p* = 0.042). In particular, mental health indicators (stress

**Table 2. Comparison of mental health scores in relation to the presence of dry eye disease symptoms in male and female.**

| Mental health | Sex | DED | | p-value |
|---|---|---|---|---|
| | | No | Yes | |
| PSS | Male | 19.39±5.62 | 20.39±5.55 | 0.531 |
| | Female | 21.38±4.68 | 24.69±4.73 | 0.020* |
| PHQ-9 | Male | 6.22±4.33 | 5.33±4.55 | 0.479 |
| | Female | 6.33±4.81 | 8.24±4.94 | 0.062 |
| GAD-7 | Male | 4.49±3.67 | 3.83±4.46 | 0.557 |
| | Female | 4.54±4.03 | 6.56±5.09 | 0.038* |

PSS, perceived stress scale; PHQ-9, patient health questionnaire-9; GAD-7, general anxiety disorder-7.

*$p < 0.050$, unpaired t-test

prominently affected the development of dry eye only in female. In the present study, stress perception, depressive symptoms, and anxiety symptoms increased the presence of dry eye by 1.28, 2.64, and 5.8 times, respectively, even after controlling for other variables in female.

Some of these studies are consistent with our findings. Previous studies have suggested that mood disorders such as depression, which are associated with eye diseases, should be considered more seriously [25, 26]. Recently, Asiedu et al. [27] demonstrated that DED symptoms affected depressive symptoms to a greater extent when other psychosomatic symptoms were present in a young population. In addition, a meta-analysis including data from 22 studies of 2.9 million patients found that anxiety and depression were more prevalent in patients with DED than in controls [28]. They reported that treatment of DED could help reduce depression symptoms, but effective management of depression could help alleviate the symptoms of DED.

Another study that evaluated dry eye and mental health problems in adult women also reported a close relationship between them [19]. Although studies have shown that women with severe psychological stress perception and depressed mood are at higher risk for developing dry eye symptoms, these studies were conducted only on adult women [19]. Many previous studies have not addressed the differences in the relationship between depression and DED between men and women.

Schaumberg et al. [13] compared the severity of DED, but only the quality of life (QoL) between men and women with DED in a large national longitudinal study targeting US citizens. Their results showed a significant difference in the Ocular Surface Disease Index (OSDI) score, which reflects the severity of DED symptoms. Women scored higher (26.8) than men (18.2) ($p < 0.001$). In addition, women are at a higher risk of being restricted from daily activities such as reading, driving at night, and watching TV which could identify their QoL with dry eye symptoms compared to men [13]. Recent clinical and epidemiologic findings also indicate that women tend to have higher risk levels of chronic pain conditions [29].

Such sex-related differences in pain are likely because of the interaction of biological differences between men and women (e.g., genetic or hormonal factors) with psychological and socio-cultural factors [29]. First, female hormones such as estrogen may be involved. Fillingim et al. [29] stated that hormonal alterations, such as estrogen administration and withdrawal, may increase pain risk. Furthermore, OSDI scores (showing eye discomfort index) were higher in the ovulation phase than in the luteal phase of the menstrual cycle [30, 31]. Secondly, the fact that DED shares genetic causes with chronic pain syndrome may be the reason [32]. The TFOS DEWS II definition and classification report has added neuropathic dry eye to the originally defined evaporative aqueous deficient dry eye classified in the dry eye symptom

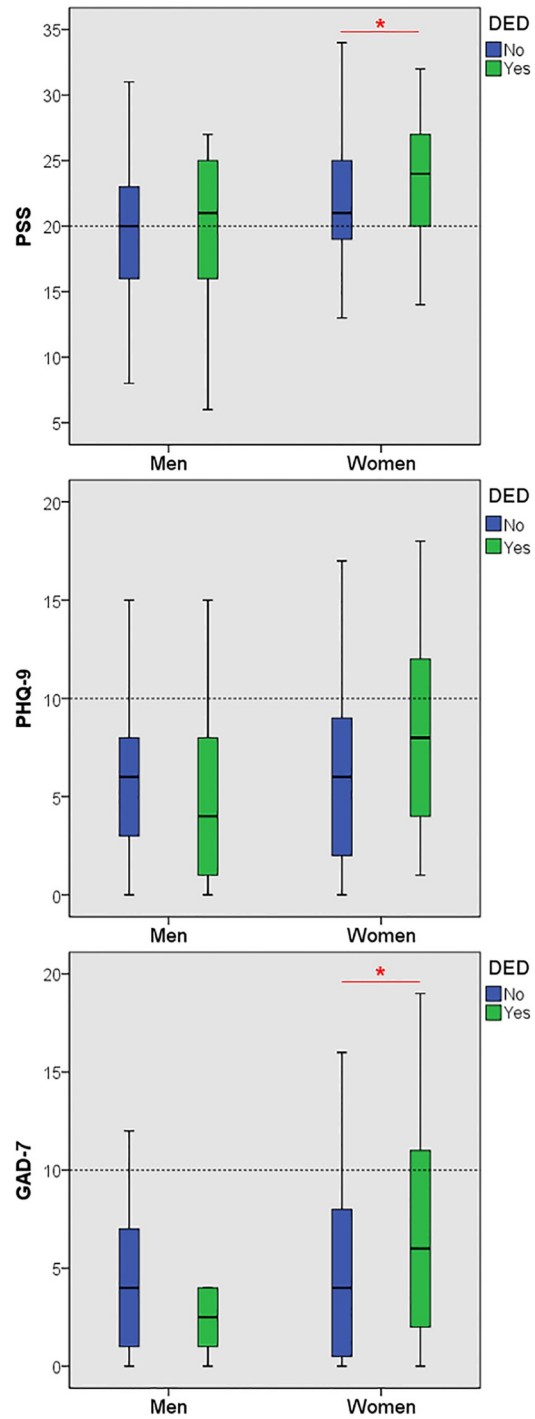

**Fig 3. The average score of mental health-related indicators, in relation to DED symptoms in male and female.**

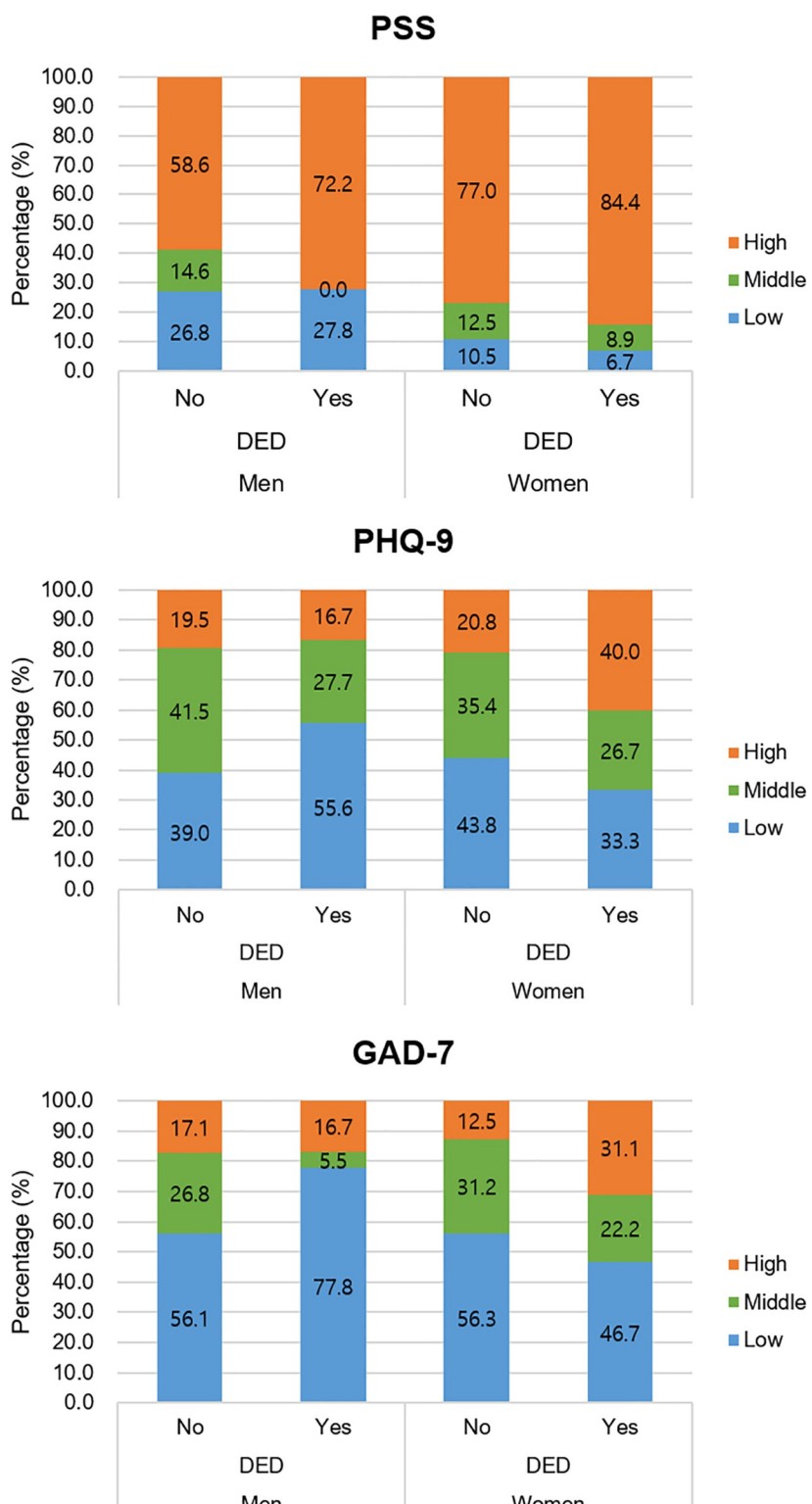

**Fig 4. The percentage of low, middle, and high score groups in mental health-related indicators, in relation to DED symptoms in male and female.**

**Table 3.  Odds ratios (OR) and 95% confidence interval(CI) on multiple logistic regression analyses of associations between DED and mental health in male and female.**

|  | Variables | Adjusted OR | (95% CI) |
|---|---|---|---|
| *Total* | Stress perception | 1.118* | (1.021–1.225) |
|  | Depressive symptom | 1.991* | (1.017–3.898) |
|  | Anxiety symptom | 3.502* | (1.093–11.223) |
| *Male* | Stress perception | 1.030 | (0.900–1.178) |
|  | Depressive symptom | 1.809 | (0.456–7.171) |
|  | Anxiety symptom | 4.808 | (0.485–47.709) |
| *Female* | Stress perception | 1.277* | (1.094–1.492) |
|  | Depressive symptom | 2.635* | (1.095–6.345) |
|  | Anxiety symptom | 5.809* | (1.156–29.144) |

Adjusted for age, sex, education, household income, marital status, occupation, body mass index, medicine, history of ocular surgery, and wearing contact lenses

*$p < 0.050$.

classification [6]. Vehof et al. [8] stated that females have a higher sensitivity to light as an environmental factor, and this is related to pain experienced in neuropathic dry eye. From these results, we can infer that there is a possibility that the frequency of neuropathic dry eye increased more in females than in males. Third, pain and depression are closely related. The prevalence of depression is higher in women than in men [33], and it is known that the sensitivity to pain and depression is accompanied by diseases [34, 35]. Therefore, we can infer that pain may cause depression, and this can be felt more acutely in women than in men [34, 35]. Additionally, men are required to be masculine socially. Slightly different from biological sex, masculinity that is required from men in social/cultural perspectives in countries/cultures/societies makes men feel that they need to be stronger than women and that they should not show their pain.

DED has been recognized as a common ocular problem affecting older women more often than men, but it should be noted that the score of mental health indicators is high in the DED patients that many women experienced in the present study. Our findings also support the suggestion of Kitazawa et al. [1] regarding dry-eyed women with depression. High stress and severe depression are strongly linked to DED; therefore, the treatment of dry eye is also effective in managing depression. However, the use of antidepressant medication can exacerbate DED, which in turn can worsen depression [36–38]. Therefore, Ayaki et al. [39] and Han et al. [14] suggested that the involvement of psychiatrists is required before managing associated sleep disorders and antidepressant use, which contributes to dry eye. We obtained the need for their suggestion to be expanded to female with dry eye and depression.

This study's major strength is that it is the first study to focus on the associations of mental health and DED symptoms with sex differences in Asians. Previous studies evaluating the relationship between DED and psychiatric or neurological disorders have not considered sex as an important factor [1, 14]. Unfortunately, little is known about dry eye problems related to mental health between men and women, although there have been studies that analyzed DED according to sex, problems related to daily life such as poor vision, reading problems, and problems watching TV were compared to sex [8].

This study had several limitations. First, the cross-sectional design did not allow us to draw conclusions about a causal relationship between DED and mental health indicators, such as stress, depression, and anxiety; thus, prospective studies are needed to determine causality.

Second, DED was classified using patient surveys, rather than objective or doctoral diagnoses. However, many cases show discordance between DED symptoms and signs, and DED symptoms regarding this study's objective are subjective and mostly scaled using surveys [22]. Third, a small number of persons participated, particularly male. However, to the best of our knowledge, our study is the first to separately evaluate DED and mental health-related indicators in male and female in the community. Future studies are also needed to determine the relationship between depression and dry eye in each age group, which can provide effective evidence for the treatment of female with both.

## Conclusion

This study contributes evidence that there is a difference in the increased risks of mental health and DED symptoms according to sex in the community. Female with DED showed higher risks of stress, depression, and anxiety than those without DED. Therefore, ophthalmologists and clinicians should be aware of the possibility of mental health problems in female patients with DED. Female adults with mental health problems may complain of developing DED. Subsequently, nurses should consider the possibility of DED in patients with mental health problems.

## Supporting information

**S1 File.**
(SAV)

## Author Contributions

**Conceptualization:** Hyojin Kim, Youngju An, Won Ju Hwang.

**Data curation:** Won Ju Hwang.

**Formal analysis:** Hyojin Kim, Youngju An.

**Funding acquisition:** Won Ju Hwang.

**Investigation:** Hyojin Kim, Youngju An, Won Ju Hwang.

**Writing – original draft:** Youngju An, Won Ju Hwang.

**Writing – review & editing:** Hyojin Kim.

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
