## [Decision Letter · Decision Letter 0]

5 Sep 2022

PONE-D-22-12671Gender Differences in Dry Eye Disease symptoms associated with Psychological Health indicators among Adults using mobile mental health apps.PLOS ONE

Dear Dr. Hwang,

Thank you for submitting your manuscript to PLOS ONE. After careful consideration, we feel that it has merit but does not fully meet PLOS ONE’s publication criteria as it currently stands. Therefore, we invite you to submit a revised version of the manuscript that addresses the points raised during the review process.

We look forward to receiving your revised manuscript.

Kind regards,

Keisuke Suzuki, MD, PhD

Academic Editor

PLOS ONE

Journal Requirements:

2. Please provide additional details regarding participant consent. In the ethics statement in the Methods and online submission information, please ensure that you have specified what type you obtained (for instance, written or verbal, and if verbal, how it was documented and witnessed). If your study included minors, state whether you obtained consent from parents or guardians. If the need for consent was waived by the ethics committee, please include this information

"No"

"No"

Reviewers' comments:

Reviewer's Responses to Questions

**Comments to the Author**

1. Is the manuscript technically sound, and do the data support the conclusions?

Reviewer #1: Yes

Reviewer #2: Yes

2. Has the statistical analysis been performed appropriately and rigorously? 

Reviewer #1: Yes

Reviewer #2: Yes

3. Have the authors made all data underlying the findings in their manuscript fully available?

Reviewer #1: Yes

Reviewer #2: Yes

4. Is the manuscript presented in an intelligible fashion and written in standard English?

Reviewer #1: Yes

Reviewer #2: Yes

5. Review Comments to the Author

Reviewer #1: The manuscript entitled “Gender Differences in Dry Eye Disease symptoms associated with Psychological Health indicators among Adults using mobile mental health apps” was an interesting research. The following comments can help the authors to improve it:

1-Please follow the journal instruction to provide a structured abstract if possible.

2-The whole manuscript needs to be revised in terms of English language.

3-The reference numbers in the main text are not sequential, and need to be corrected.

4-Abbreviations need to be presented in their full form in the first place that they are used.

5-The introduction section is so brief and the gap of knowledge needs to be explained with more details.

6- More information about the App needs to be added to the manuscript.

7-More information about the settings of the study and the reasons for excluding patients aged 61 or more need to be added.

8-As the number of women was higher than men, is there any possibility for bias in reporting the results? Why didn’t the researchers recruit the same number of patients for males and females?

Reviewer #2: In Result part:

1. Line 94-95: The first sentence was wrong. Please remove.

2. It would be better result if the author can analyses more in each age because as we know that dry eye mostly happens in aging female.

6. PLOS authors have the option to publish the peer review history of their article (what does this mean?). If published, this will include your full peer review and any attached files.

Reviewer #1: No

Reviewer #2: **Yes: **Chulaluck Tangmonkongvoragul

---

## [Author Response · Author response to Decision Letter 0]

8 Oct 2022

Dear editor:

We would like to thank you and the reviewers of the Plos One for taking the time to review our article. We revised the manuscript, based on your comments. The changes are summarized below: 

Reviewer #1: 

Comment 

The manuscript entitled “Gender Differences in Dry Eye Disease symptoms associated with Psychological Health indicators among Adults using mobile mental health apps” was an interesting research. The following comments can help the authors to improve it:

1-Please follow the journal instruction to provide a structured abstract if possible. [Response] Thank you for your comments. We revised it to be a structured abstract.

2-The whole manuscript needs to be revised in terms of English language. 

[Response] Thank you for your comments. The manuscript was edited by English speaker.

3-The reference numbers in the main text are not sequential, and need to be corrected. [Response] Thank you for your comments. We confirmed the reference numbers again. Also, 6 references (number of 9, 10, 11, 12, 16, 17) have been added.

4-Abbreviations need to be presented in their full form in the first place that they are used. 

[Response] Thank you for your comments. We have corrected the abbreviations to the submission format.

5-The introduction section is so brief and the gap of knowledge needs to be explained with more details. 

[Response] Thank you for your comments. We added the introduction section. (Page 4, lines 61, 66-71)

6- More information about the App needs to be added to the manuscript. 

[Response] Thank you for your comments. We added the App information in method section. (Page 4, lines 87-89)

7. More information about the settings of the study and the reasons for excluding patients aged 61 or more need to be added. 

[Response] We agree with your comments. We added the reasons for excluding patients aged 61 or more and references. (Page 5, lines 95-98) 

The previous study has demonstrated that over 60 age was a significant risk factor for dry eye symptoms. Therefore, older participants over 60 were not included. Also, we revised the manuscript to be exactly 60 years old. In addition, because it was reported that those over the age of 60 who used health apps had low self-efficacy, we excluded those over the age of 60. 

8-As the number of women was higher than men, is there any possibility for bias in reporting the results? Why didn’t the researchers recruit the same number of patients for males and females? 

[Response] Thank you for your comments. 

There were more women who participated to use the health app. Therefore, we did not adjust to match the number of men and women in our statistical analysis. Since we showed the analysis results for men and women separately in Tables 1, 2, and 3, we think that it would not affected the results.

Reviewer #2: 

Comment 

In Result part:

1. Line 94-95: The first sentence was wrong. Please remove.

[Response] Thank you for your comments. We removed the sentence. 

2. It would be better result if the author can analyses more in each age because as we know that dry eye mostly happens in aging female.

[Response] Many thanks for your comments. 

We considered dividing the analysis into each age groups, but we did not include such an analysis because we excluded the old female group by adding to the reasons for excluding the age of 60 or older in the method. In addition, these sentences were described to the Discussion section. (page 16, line 303)

Sincerely yours,

---

## [Decision Letter · Decision Letter 1]

17 Oct 2022

PONE-D-22-12671R1Gender Differences in Dry Eye Disease symptoms associated with Psychological Health indicators among Adults using mobile mental health apps.PLOS ONE

Dear Dr. Hwang,

Thank you for submitting your manuscript to PLOS ONE. After careful consideration, we feel that it has merit but does not fully meet PLOS ONE’s publication criteria as it currently stands. Therefore, we invite you to submit a revised version of the manuscript that addresses the points raised during the review process.

We look forward to receiving your revised manuscript.

Kind regards,

Keisuke Suzuki, MD, PhD

Academic Editor

PLOS ONE

Journal Requirements:

Additional Editor Comments (if provided):

The manuscript has been improved, but please make further minor revisions to the manuscript in accordance with the reviewers' comments.

Reviewers' comments:

Reviewer's Responses to Questions

**Comments to the Author**

1. If the authors have adequately addressed your comments raised in a previous round of review and you feel that this manuscript is now acceptable for publication, you may indicate that here to bypass the “Comments to the Author” section, enter your conflict of interest statement in the “Confidential to Editor” section, and submit your "Accept" recommendation.

Reviewer #1: (No Response)

2. Is the manuscript technically sound, and do the data support the conclusions?

Reviewer #1: Yes

3. Has the statistical analysis been performed appropriately and rigorously? 

Reviewer #1: Yes

4. Have the authors made all data underlying the findings in their manuscript fully available?

Reviewer #1: Yes

5. Is the manuscript presented in an intelligible fashion and written in standard English?

Reviewer #1: No

6. Review Comments to the Author

Reviewer #1: I appreciate the authors for their time and efforts to revise the manuscript. However, it still needs to be improved in the following areas:

1- Again, the whole manuscript needs to be revised in terms of English language. Moreover, please be consistent when using (.), (-) and (,).

2- To report the results, we unusually use “sex” which is categorized to “male” and “female”. Please consider these two in the whole manuscript.

3- In the introduction section, the terms sex and gender were used next to each other. Please only use one term "sex".

4- Please revise the first paragraph of the methods section, if you are explaining inclusion and exclusion criteria in other paragraphs.

5- More information about the App needs to be added to the manuscript. Adding only two lines to the methods section is not adequate. Please add some figures of the App, too.

6- More information about the settings of the study needs to be added.

7- In the tables, there are a number of cells which show no label or figure. Please complete the tables or merge cells, etc.

7. PLOS authors have the option to publish the peer review history of their article (what does this mean?). If published, this will include your full peer review and any attached files.

Reviewer #1: No

---

## [Author Response · Author response to Decision Letter 1]

28 Oct 2022

I appreciate the authors for their time and efforts to revise the manuscript. However, it still needs to be improved in the following areas:

1- Again, the whole manuscript needs to be revised in terms of English language. Moreover, please be consistent when using (.), (-) and (,).

→Answer: We have revised all the marks to be consistent throughout the paper. Also, the manuscript was edited by an English native speaker. 

2- To report the results, we unusually use “sex” which is categorized to “male” and “female”. Please consider these two in the whole manuscript.

→Answer: We have revised all “men / women” to “male / female” throughout the paper.

3- In the introduction section, the terms sex and gender were used next to each other. Please only use one term "sex".

→Answer: We have revised all “gender” to “sex” throughout the paper.

4- Please revise the first paragraph of the methods section, if you are explaining inclusion and exclusion criteria in other paragraphs.

→Answer: We have consolidated the paragraphs and modified them to fit the context.

5- More information about the App needs to be added to the manuscript. Adding only two lines to the methods section is not adequate. Please add some figures of the App, too.

→Answer: We have added some figures for improvement.

6- More information about the settings of the study needs to be added.

→Answer: As a primary study, this study was first aimed at examining the participants' mental health and dry eye symptoms. In the future, we plan to have participants use the healing program on a regular basis through a mobile app to investigate the intervention effect. 

7- In the tables, there are a number of cells which show no label or figure. Please complete the tables or merge cells, etc.

→Answer: We follow the reviewer’s suggestion.

Best regards

---

## [Editor Report · Decision Letter 2]

24 Nov 2022

Gender Differences in Dry Eye Disease symptoms associated with Psychological Health indicators among Adults using mobile mental health apps.

PONE-D-22-12671R2

Dear Dr. Hwang,

We’re pleased to inform you that your manuscript has been judged scientifically suitable for publication and will be formally accepted for publication once it meets all outstanding technical requirements.

Kind regards,

Keisuke Suzuki, MD, PhD

Academic Editor

PLOS ONE

Additional Editor Comments (optional):

The authors have well addressed comments raised by reviewers.
---

## [Editor Report · Acceptance letter]

5 Jan 2023

PONE-D-22-12671R2 

Gender differences in dry eye disease symptoms associated with psychological health indicators among adults using mobile mental health apps 

Dear Dr. Hwang:

I'm pleased to inform you that your manuscript has been deemed suitable for publication in PLOS ONE. Congratulations! Your manuscript is now with our production department. 

Kind regards, 

on behalf of

Dr. Keisuke Suzuki 

Academic Editor

PLOS ONE